# Factors associated with COVID-19 vaccine uptake and hesitancy among healthcare workers in the Democratic Republic of the Congo

**Michel K. Nzaji**[1], **Jean de Dieu Kamenga**[1], **Christophe Luhata Lungayo**[2], **Aime Cikomola Mwana Bene**[2], **Shanice Fezeu Meyou**[3], **Anselme Manyong Kapit**[1], **Alanna S. Fogarty**[4], **Dana Sessoms**[3], **Pia D. M. MacDonald**[3,5], **Claire J. Standley**[4,6]*, **Kristen B. Stolka**[3]

**1** Social, Statistical and Environmental Sciences, RTI International, Kinshasa, Democratic Republic of Congo, **2** Expanded Programme on Immunization, Ministry of Public Health, Kinshasa, Democratic Republic of Congo, **3** Social, Statistical and Environmental Sciences, RTI International, Research Triangle Park, North Carolina, United States of America, **4** Center for Global Health Science and Security, Georgetown University, Washington, District of Columbia, United States of America, **5** Department of Epidemiology, Gillings School of Global Public Health, University of North Carolina, Chapel Hill, North Carolina, United States of America, **6** Heidelberg Institute of Global Health, University of Heidelberg, Heidelberg, Germany

* Claire.standley@georgetown.edu

## Abstract

Vaccination is a critical intervention to reduce morbidity and mortality and limit strain on health systems caused by COVID-19. The slow pace of COVID-19 vaccination uptake observed in some settings raises concerns about COVID-19 vaccine hesitancy. The Democratic Republic of the Congo experienced logistical challenges and low uptake at the start of vaccine distribution, leading to one of the lowest overall COVID-19 vaccine coverage rates in the world in 2021. This study assessed the magnitude and associated factors of COVID-19 vaccine uptake among healthcare workers (HCWs) in seven provinces in DRC. We implemented a cross-sectional Knowledge, Attitudes, and Practices (KAP) questionnaire targeting HCWs, administered by trained data collectors in Haut-Katanga, Kasaï Orientale, Kinshasa, Kongo Centrale, Lualaba, North Kivu, and South Kivu provinces. Data were summarized and statistical tests were performed to assess factors associated with vaccine uptake. HCWs across the seven provinces completed the questionnaire (N = 5,102), of whom 46.3% had received at least one dose of COVID-19 vaccine. Older age, being married, being a medical doctor, being a rural resident, and having access to or having previously worked in a COVID-19 vaccination site were all strongly associated with vaccination uptake. Vaccinated individuals most frequently cited protection of themselves, their families, and their communities as motivations for being vaccinated, whereas unvaccinated individuals were most concerned about safety, effectiveness, and risk of severe side effects. The findings suggest an opinion divide between vaccine-willing and vaccine-hesitant HCWs. A multidimensional approach may be needed to increase the acceptability of the COVID-19 vaccine for HCWs. Future vaccine campaign messaging could center around the positive impact of vaccination on protecting friends, family, and the community, and also emphasize

**Data Availability Statement:** All underlying data are available in the manuscript or supporting materials.

**Funding:** This publication was supported by funding from Cooperative Agreement NU2HGH000047 funded by the US Centers for Disease Control and Prevention. Individuals from the US CDC provided technical input into the design of the study but were not involved in the analysis or preparation of the manuscript.

**Competing interests:** The authors have declared that no competing interests exist.

the safety and very low risk of adverse effects. These types of messages may further be useful when planning future immunization campaigns with new vaccines.

## Introduction

Coronavirus disease 2019 (COVID-19), which is caused by the novel severe acute respiratory syndrome coronavirus 2 (SARS-CoV-2), was first recognized in late 2019 and declared a global pandemic by the World Health Organization in March 2020 [1]. While SARS-CoV-2 can result in serious complications[2], COVID-19 vaccines have been shown to be effective in preventing severe disease [3]. Real-world data have also shown that COVID-19 vaccines reduced the risk of COVID-19–associated deaths, regardless of the emergence of the Delta and the Omicron variants [4]. The speed of development and production of the COVID-19 vaccine is unprecedented; however, some data suggest this could contribute to poorer perceptions of the vaccine's efficacy and safety [5].

Healthcare workers (HCWs), defined as any individual who directly or indirectly delivers care or services to the sick [6], are at high risk of occupational exposure to and transmission of SARS-CoV-2, which prioritized them for early vaccination against COVID-19 [7]. HCWs also play an important role in immunization programs because they not only administer vaccines but they also educate, influence, and build trust with patients around vaccination [8]. In this way, communities treat HCWs as role models for their attitudes toward vaccination and refer to them for vaccine information [9]. Consequently, vaccine uptake among HCWs may encourage widespread uptake in vaccination among the general population. Conversely, if HCWs are hesitant to be vaccinated, it can be directly detrimental to the response effort if they suffer higher rates of infection and morbidity and this, in turn, can influence negative vaccine perceptions in the public. Thus, assessing the factors and reasons associated with HCW uptake and hesitancy is important to help inform targeted approaches for reducing vaccine hesitancy and increasing confidence in vaccines.

As of 23 April 2023, the Democratic Republic of the Congo (DRC) has reported just under 96,000 confirmed COVID-19 cases and 1,465 COVID-19–related deaths since the beginning of the pandemic. These are likely substantial underestimates of the true impact of COVID-19 in DRC, given low testing rates and observed high test positivity rates across successive epidemiological waves [10]; estimates based on excess mortality calculations suggest a much higher fatality rate than reported [11].

DRC has one of the lowest rates of COVID-19 vaccine coverage in the world, with only 15.5% of the population having received at least one dose by April 2023 [12]. The vaccine campaign in DRC was also slow to get underway; by the end of 2021, fewer than 1% of the population had received a single dose of the vaccine. By contrast, almost 80% of people in Vietnam (similar total population size to DRC), 19% of people in Liberia (similar gross domestic product per capita as DRC), and 16% of people in Algeria (similar land area to DRC) had received at least one dose of COVID-19 vaccine by this time [13]. While the vaccine rollout in DRC was hindered by operational and logistical factors, including availability of doses [14], previous studies, which were mostly conducted in single locations or with relatively small samples, have demonstrated that vaccine hesitancy was also a factor [15,16]. Consequently, this study aimed to assess the magnitude and associated factors of COVID-19 vaccine uptake and hesitancy among a large number of HCWs across seven DRC provinces.

## Methods

### Study area

Between 24 December 2021 and 1 March 2023, a Knowledge, Attitudes, and Practices (KAP) questionnaire was administered to HCWs in seven DRC provinces: Haut-Katanga, Kasaï Orientale, Kinshasa, Kongo Centrale, Lualaba, North Kivu, and South Kivu (Fig 1). Provinces were selected as part of an effort to implement intra-action reviews (IARs) in priority provinces. The questionnaires were administered in the 2 weeks prior to the IAR to contribute to learning and sharing of best practices and challenges around COVID-19 vaccination at the provincial level [17,18].

### Study population

Public health facilities and private hospitals located in and around the capital cities of the seven targeted provinces were selected through convenience sampling and the questionnaires

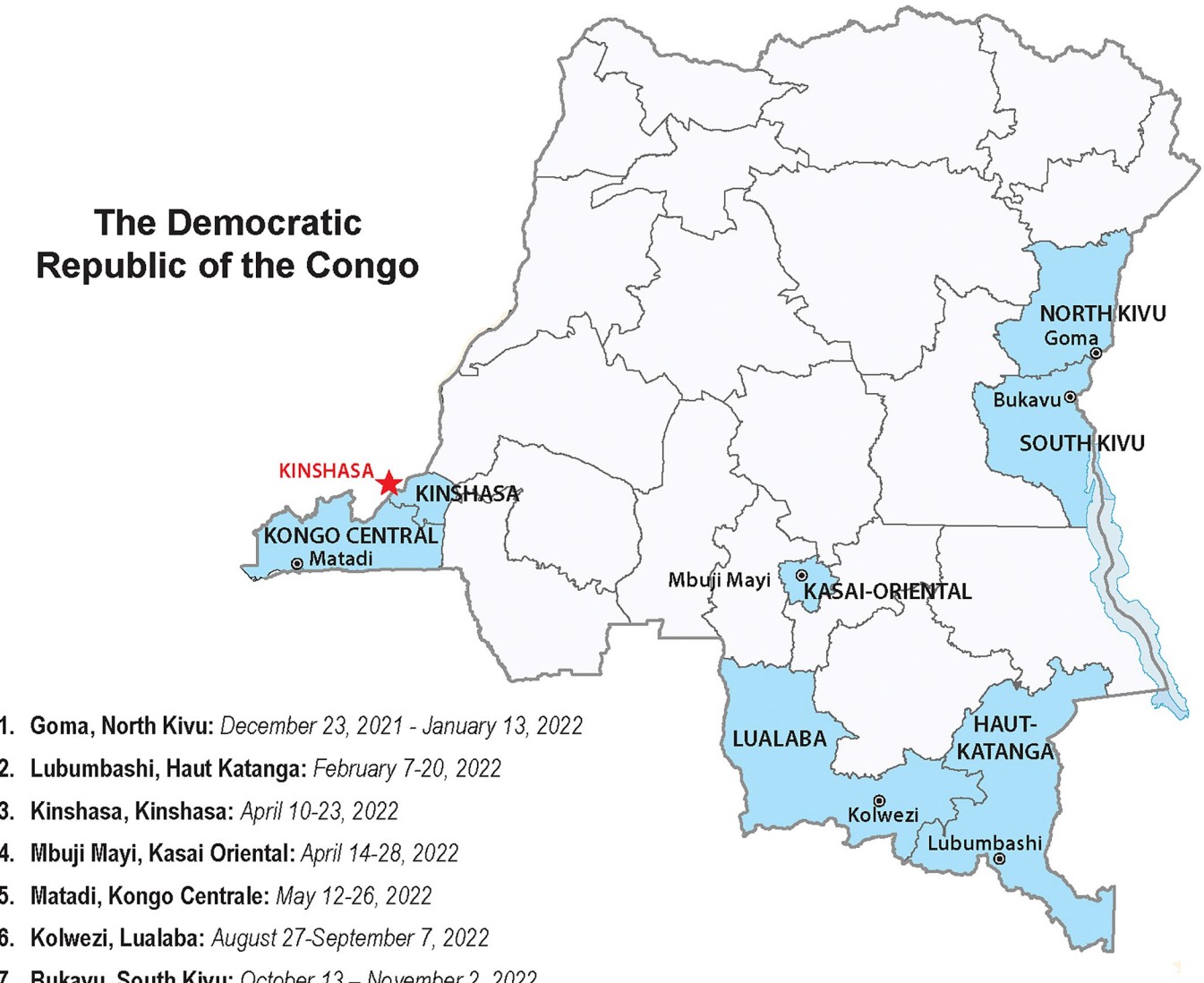

1. **Goma, North Kivu:** *December 23, 2021 - January 13, 2022*

2. **Lubumbashi, Haut Katanga:** *February 7-20, 2022*

3. **Kinshasa, Kinshasa:** *April 10-23, 2022*

4. **Mbuji Mayi, Kasai Oriental:** *April 14-28, 2022*

5. **Matadi, Kongo Centrale:** *May 12-26, 2022*

6. **Kolwezi, Lualaba:** *August 27-September 7, 2022*

7. **Bukavu, South Kivu:** *October 13 – November 2, 2022*

**Fig 1. Map of seven provinces and dates when KAP questionnaire was administered in DRC, December 2021-March 2023.** Figure prepared using base layer from the CIA World Factbook (https://www.cia.gov/static/b2fcc8d80f910b0c91f4a74d33b5c7e6/DRC_Administrative.pdf).

were administered widely to all available HCWs working in these workplaces. Participating HCWs included doctors, nurses, midwives, laboratory technicians, administrative personnel, and others, aged 18 or older, who provided informed consent.

The Cochrane formula [19], where n equals minimum sample size, Z represents the standard normal deviate corresponding to 5% significant level, and p equals proportion of HCWs who are COVID-19 vaccine hesitant, was used to estimate the target sample size per province. Because we did not find a reference study on vaccine hesitancy in DRC at the time of the study design, we estimated 50% of HCWs to be hesitant to vaccination against COVID-19, d = tolerable error of margin set at 0.05; therefore, Z = 1.96. A confidence level of 95% and a margin of error of 5% were used and resulted in a minimum sample size of 484 participants per province after accounting for a nonresponse rate or 10% incomplete response.

### Data collection and analysis

The questionnaire consisted of questions that assessed demographics, health history, COVID-19 vaccine uptake (at least one dose), perception of risk and exposure to COVID-19, confidence in the COVID-19 response, stated reasons for acceptance or rejection of the COVID-19 vaccine, exposure to information about COVID-19, and intention to vaccinate. The questions —and for some questions, response options—in the questionnaire were derived from the literature on vaccine hesitancy and acceptability (S1 File) [20,21]. Each trained data collector conducted a pretest of the questionnaire tool with 10 HCWs and convened after the pretest to provide feedback on their experience. The final questionnaire was administered to HCWs by trained data collectors and the data were entered electronically into the questionnaire programmed in KoboCollect (https://www.kobotoolbox.org/).

Completed questionnaires were exported from KoboCollect to Microsoft Excel for cleaning and coding. Responses were analyzed using SPSS Enterprise Guide Version 22, with verification of results, and calculation of confidence intervals, performed in StatCal (EpiInfo 7). Associations between independent variables and the primary outcomes (vaccinated or not vaccinated) were tested using Student's t-tests or chi-square tests, as appropriate. Student's t-tests and ANOVA were used to test for differences between means of Likert scale variables. Nonbinary variables were dichotomized against the reference variable and a step-by-step, bottom-up Wald analysis was performed to define the variables to be included in the final multivariable logistic regression model. The p-value was set at alpha = 0.05 for significance testing.

### Ethical statement

The study was approved by the ethics committee of the School of Public Health at the University of Lubumbashi, DRC (approval letter No UNILU/CEM/104/2022). All study participants provided verbal informed consent prior to completing the questionnaire. Documentation for verbal consent was not required due to the one-off nature of the study (no follow-up with participants) and as the methods represented minimal risk to the subjects.

## Results

Overall, 5,102 individuals provided responses to the questionnaire, of whom 832 were in Haut Katanga, 550 in Kasai Oriental, 900 in Kinshasa, 896 in Kongo Central, 591 in Lualaba, 422 in North Kivu, and 911 in South Kivu. The full dataset of coded responses is provided in the Supplemental Material (S2 File). Sociodemographic characteristics of the respondents, by province, are provided in Table 1.

Overall, 46.3% of respondents reported having received one or more doses of the COVID-19 vaccine, but with substantial variation between provinces (Table 2). Three-quarters of the

**Table 1. Sociodemographic characteristics of participants, by DRC province (N = 5,102 except where otherwise indicated).**

| | Haut Katanga (%) | Kasai Oriental (%) | Kinshasa (%) | Kongo Central (%) | Lualaba (%) | North Kivu (%) | South Kivu (%) | Total (%) |
|---|---|---|---|---|---|---|---|---|
| **Age range** | | | | | | | | |
| 18–29 | 162 (19.5%) | 103 (18.7%) | 78 (8.7%) | 61 (6.8%) | 152 (25.7%) | 128 (30.3%) | 197 (21.6%) | **881 (17.3%)** |
| 30–39 | 274 (32.9%) | 162 (29.5%) | 362 (40.2%) | 258 (28.8%) | 200 (33.8%) | 159 (37.7%) | 325 (35.7%) | **1,740 (34.1%)** |
| 40–54 | 285 (32.3%) | 205 (37.3%) | 374 (41.6%) | 474 (52.9%) | 218 (36.9%) | 120 (28.4%) | 294 (32.3%) | **1,970 (38.6%)** |
| >55 | 111 (14.3%) | 80 (14.5%) | 86 (9.6%) | 103 (11.5%) | 21 (3.6%) | 15 (3.6%) | 95 (10.4%) | **511 (10.0%)** |
| Provincial n | 832 | 550 | 900 | 896 | 591 | 422 | 911 | **5,102** |
| **Sex** | | | | | | | | |
| Female | 421 (50.6%) | 272 (49.5%) | 453 (50.3%) | 534 (59.6%) | 314 (53.1%) | 179 (42.4%) | 392 (43.0%) | **2,565 (50.3%)** |
| Male | 411 (48.4%) | 278 (50.5%) | 447 (49.7%) | 362 (40.4%) | 277 (46.9%) | 243 (57.6%) | 519 (57.0%) | **2,537 (49.7%)** |
| **Level of education** | | | | | | | | |
| None | 16 (1.9%) | 1 (0.2%) | 4 (0.4%) | 14 (1.6%) | 12 (2.0%) | 8 (1.9%) | 5 (0.5%) | **60 (1.2%)** |
| Elementary school | 30 (3.6%) | 40 (7.3%) | 6 (0.7%) | 24 (2.7%) | 11 (1.9%) | 16 (3.8%) | 33 (3.6%) | **160 (3.0%)** |
| Middle school | 171 (20.6%) | 200 (36.4%) | 108 (12.0%) | 305 (34.0%) | 59 (10.0%) | 79 (18.7%) | 191 (21.0%) | **1,113 (21.8%)** |
| University or higher | 615 (73.9%) | 309 (56.2%) | 782 (86.9%) | 553 (61.7%) | 509 (86.1%) | 319 (75.6%) | 682 (74.9%) | **3,769 (73.9%)** |
| **Healthcare worker categories** | | | | | | | | |
| Nurse | 364 (43.8%) | 268 (48.7%) | 398 (44.2%) | 458 (51.1%) | 270 (45.7%) | 165 (39.1%) | 433 (47.5%) | **2,356 (46.2%)** |
| Doctor | 195 (23.4%) | 78 (14.2%) | 256 (28.4%) | 87 (9.7%) | 47 (8.2%) | 73 (17.3%) | 78 (8.6%) | **814 (16.0%)** |
| Pharmacist | 24 (2.9%) | 10 (1.8%) | 30 (3.3%) | 11 (1.2%) | 62 (10.5%) | 19 (4.5%) | 52 (5.7%) | **208 (4.1%)** |
| Midwife | 47 (5.6%) | 25 (4.5%) | 71 (7.9%) | 62 (6.9%) | 86 (14.6%) | 23 (5.5%) | 130 (14.3%) | **444 (8.7%)** |
| Laboratory technician | 67 (8.1%) | 50 (9.1%) | 70 (7.8%) | 128 (14.3%) | 50 (8.5%) | 19 (4.5%) | 78 (8.6%) | **462 (9.1%)** |
| Other | 135 (16.2%) | 119 (21.6%) | 75 (8.3%) | 150 (16.6%) | 76 (12.9%) | 123 (29.1%) | 140 (15.4%) | **818 (16.0%)** |
| **Marital status** | | | | | | | | |
| Single | 126 (15.1%) | 52 (9.5%) | 129 (14.3%) | 119 (13.3%) | 182 (30.8%) | 121 (28.7%) | 197 (21.6%) | **926 (18.1%)** |
| Divorced/ Separated | 15 (1.8%) | 9 (1.6%) | 13 (1.4%) | 35 (3.9%) | 7 (1.2%) | 2 (0.5%) | 9 (1.0%) | **90 (1.8%)** |
| Married | 649 (78.0%) | 463 (84.2%) | 539 (59.9%) | 560 (62.5%) | 360 (60.9%) | 284 (67.3%) | 685 (75.2%) | **3,540 (69.4%)** |
| Cohabitation | 13 (1.6%) | 1 (0.2%) | 198 (22.0%) | 148 (16.5%) | 2 (0.5%) | 11 (2.6%) | 3 (0.3%) | **376 (7.4%)** |
| Widowed | 29 (3.5%) | 25 (4.5%) | 21 (2.3%) | 34 (3.8%) | 40 (6.8%) | 4 (0.9%) | 17 (1.9%) | **170 (3.3%)** |
| **Religion** | | | | | | | | |
| Animist | 4 (0.5%) | 2 (0.4%) | 2 (0.2%) | 16 (1.8%) | 2 (0.3%) | 2 (0.5%) | 0 (0%) | 28 (0.5%) |

*(Continued)*

**Table 1.** (Continued)

| | Haut Katanga (%) | Kasai Oriental (%) | Kinshasa (%) | Kongo Central (%) | Lualaba (%) | North Kivu (%) | South Kivu (%) | Total (%) |
|---|---|---|---|---|---|---|---|---|
| Christian | 793 (95,3%) | 522 (94,9%) | 879 (97,7%) | 836 (93,3%) | 537 (90,9%) | 402 (95,3%) | 892 (97,9%) | 4861 (95,3%) |
| Muslim | 27 (3,2%) | 9 (1,6%) | 9 (1,0%) | 3 (0,3%) | 13 (2,2%) | 6 (1,4%) | 9 (1,0%) | 76 (1,5%) |
| Without religion | 8 (1,0%) | 12 (2,2%) | 4 (0,4%) | 6 (0,7%) | 17 (2,9%) | 4 (0,9%) | 7 (0,8%) | **58 (1,1%)** |
| Other | 0 (0%) | 5 (0,9%) | 6 (0,7%) | 35 (3,9%) | 22 (3,7%) | 8 (1,9%) | 3 (0,3%) | **79 (1,5%)** |
| **Place of residence** | | | | | | | | |
| Urban | 773 (92.9%) | 470 (85.5%) | 898 (99.8%) | 896 (100%) | 526 (89.0%) | 330 (78.2%) | 595 (65.3%) | **4488 (88.0%)** |
| Rural | 59 (7.1%) | 80 (14.5%) | 2 (0.2%) | 0 (0.00%) | 65 (11.0%) | 92 (21.8%) | 316 (34.7%) | **614 (12.0%)** |
| **Other vaccine uptake, excluding routine childhood immunizations** | | | | | | | | |
| Yes | 152 (18.3%) | 106 (19.3%) | 412 (45. 8%) | 476 (53.1%) | 91 (15.4%) | 305 (72.3%) | 423 (46.4%) | **1965 (38.53%)** |
| No | 680 (81.7%) | 444 (80.7%) | 488 (54.2%) | 420 (46.9%) | 500 (84.6%) | 117 (27.7%) | 488 (53. 6%) | **3137 (61.5%)** |
| **Types of other vaccines taken (N = 1965)** | | | | | | | | |
| Cholera | 90 (50.2%) | 66 (60.0%) | 13 (2.9%) | 1 (0.2%) | 2 (2.2%) | 105 (21.9%) | 156 (28.5%) | **433 (18.4%)** |
| Ebola | 3 (1.6%) | 2 (1.8%) | 30 (6.8%) | 1 (0.2%) | 1 (1.1%) | 256 (53.4%) | 197 (35.9%) | **490 (20.8%)** |
| Yellow fever | 69 (38.5%) | 9 (8.2%) | 377 (85. 1%) | 463 (84. 7%) | 83 (94.6%) | 45 (9.4%) | 76 (13.9%) | **1122 (47.6%)** |
| Meningitis | 4 (2.2%) | 0 (0.00%) | 0 (0.00%) | 0 (0.00%) | 1 (1.3%) | 50 (10.4%) | 22 (4.0%) | **77 (3.3%)** |
| Tetanus | 8 (4.5%) | 29 (26.4%) | 21 (4.7%) | 32 (5.8%) | 5 (5.4%) | 14 (2.9%) | 70 (12.8%) | **179 (7.6%)** |
| Other | 5 (2.8%) | 4 (3.7%) | 2 (0.5%) | 6 (1.1%) | 0 (0.00%) | 9 (1.8%) | 27 (3.1%) | **53 (2.3%)** |
| **Existing chronic illness** | | | | | | | | |
| Yes | 111 (13.3%) | 85 (15.5%) | 157 (17.4%) | 63 (7.0%) | 68 (11.5%) | 46 (10.9%) | 32 (3.5%) | **562 (11.0%)** |
| No | 693 (83.3%) | 456 (82.9%) | 683 (75. 9%) | 809 (90.3%) | 510 (86.3%) | 357 (84.6%) | 856 (94.0%) | **4364 (85.5%)** |
| I don't know | 28 (3.4%) | 9 (1.6%) | 60 (6.7%) | 24 (2. 7%) | 13 (2.2%) | 19 (4.5%) | 23 (2.5%) | **176 (3.4%)** |

Percentages are calculated across rows. Reference variable noted in the OR column. OR = odds ratio; CI = confidence interval.

respondents believed they were at either moderate or high risk with respect to contracting COVID-19, although less than 44% had ever been tested for COVID-19, and only about a third reported having been in contact with a COVID-19 patient.

The multivariable logistic regression suggested that several factors were significantly associated with receiving at least one dose of COVID-19 vaccine (Table 3). Being in the older age group (55 years or older), being a doctor (compared with all other types of HCWs), being married, and being a rural resident were all associated with being vaccinated, as was having received other adult non-COVID-19 vaccinations. Other significant factors for respondents related to having access to vaccination through their work or within their health structure or having knowledge about vaccination efforts.

**Table 2. COVID-19 beliefs and practices, by province (N = 5,102).**

| | Haut Katanga (%) | Kasai Oriental (%) | Kinshasa (%) | Kongo Central (%) | Lualaba (%) | North Kivu (%) | South Kivu (%) | Total (%) |
|---|---|---|---|---|---|---|---|---|
| **Are you vaccinated against COVID-19?** | | | | | | | | |
| Yes–at least one dose | 355 (42.7%) | 408 (70.8%) | 382 (42.4%) | 347 (38.7%) | 268 (45.3%) | 138 (32.7%) | 466 (51.2%) | **2364 (46.3%)** |
| No | 477 (57.3%) | 142 (25.8%) | 518 (57.6%) | 549 (61.3%) | 323 (54.7%) | 284 (67.3%) | 445 (48.8%) | **2738 (53.7%)** |
| **What is your risk of contracting COVID-19?** | | | | | | | | |
| No risk | 17 (2.0%) | 97 (17.6%) | 22 (2.4%) | 119 (13.3%) | 71 (12.0%) | 18 (4.3%) | 24 (2.6%) | **368 (7.2%)** |
| Low | 125 (15.0%) | 80 (14.5%) | 111 (12.3%) | 144 (16.1) | 222 (37.6%) | 73 (17. 3%) | 107 (11.7%) | **862 (16.9%)** |
| Moderate | 421 (50.6%) | 149 (27.1%) | 525 (58.3%) | 412 (46.0%) | 189 (32.0%) | 111 (26.3%) | 342 (37.5%) | **2149 (42.1%)** |
| High | 269 (32.3%) | 224 (40.7%) | 242 (26.9%) | 221 (24.7%) | 109 (18. 4%) | 220 (52. 1%) | 438 (48.1%) | **1723 (33.8%)** |
| **Have you ever been tested for COVID-19?** | | | | | | | | |
| Yes | 431 (51.8%) | 107 (19.5%) | 494 (54.5%) | 423 (47.2%) | 204 (34.5%) | 184 (43.6%) | 373 (40.9%) | **2216 (43.4%)** |
| No | 401 (48.2%) | 443 (80.5%) | 406 (45.1%) | 473 (52.8%) | 387 (65.5%) | 238 (56.4%) | 538 (59.1%) | **2886 (56.6%)** |
| **Knowledge of availability of different COVID-19 vaccines in province** | | | | | | | | |
| Yes | 775 (93.1%) | 480 (87.3%) | 839 (93,2%) | 740 (82.6%) | 568 (96.1%) | 358 (84.8%) | 817 (89.7%) | **4577 (89.7%)** |
| No | 156 (6.9%) | 70 (12.7%) | 61 (6.8%) | 156 (17.4%) | 23 (3.9%) | 64 (15.2%) | 94 (10.3%) | **525 (10.3%)** |
| **Awareness of routine vaccination against COVID-19 in province or local area** | | | | | | | | |
| Yes | 768 (92.3%) | 348 (63.3%) | 791 (87.9%) | 683 (76.2%) | 571 (96.6%) | 191 (45.3%) | 811 (89.0%) | **4163 (81.59%)** |
| No | 64 (7.7%) | 202 (36.7%) | 109 (12.1%) | 213 (23.8%) | 20 (3.4%) | 231 (54.7%) | 100 (11.0%) | **939 (18.4%)** |
| **Aware of the planned vaccination campaign against COVID-19 in province or local area** | | | | | | | | |
| Yes | 678 (81.5%) | 502 (91.3%) | 878 (97.6%) | 879 (98.1%) | 517 (87.5%) | 392 (92.9%) | 899 (98.7%) | **4745 (93.0%)** |
| No | 154 (18.5%) | 48 (8.7%) | 22 (2.4%) | 17 (1.9%) | 74 (12.5%) | 30 (7.1%) | 12 (1.3%) | **357 (7.0%)** |
| **Vaccination within respondent's facility** | | | | | | | | |
| Yes | 754 (90.6%) | 367 (66.7%) | 796 (88.4%) | 467 (52.1%) | 388 (65.7%) | 422 (100.0%) | 633 (69.5%) | **3827 (75.0%)** |
| No | 78 (9.4%) | 183 (33.3%) | 104 (11.6%) | 429 (47.9%) | 203 (34.3%) | 0 (0.0%) | 278 (30.5%) | **1275 (25.0%)** |
| **Previous work at a COVID-19 vaccination site** | | | | | | | | |
| Yes | 166 (20.0%) | 122 (22.2%) | 275 (30.6%) | 123 (13.7%) | 162 (27.4%) | 422 (100.0%) | 270 (29.6%) | **1540 (30.2%)** |
| No | 666 (80.0%) | 428 (77.8%) | 625 (69.4%) | 773 (86.3%) | 429 (72.6%) | 0 (0.0%) | 641 (70.4%) | **3562 (69.8%)** |
| **Willingness to take a COVID-19 vaccination if available in the province** | | | | | | | | |
| Yes | 477 (57.3%) | 473 (86.0%) | 563 (62.6%) | 555 (61.9%) | 490 (82.9%) | 246 (58.3%) | 626 (68.7%) | **3430 (67.2%)** |
| No | 355 (42.7%) | 77 (14.0%) | 337 (37.4%) | 341 (38.1%) | 101 (17.1%) | 176 (41.7%) | 285 (31.3%) | **1672 (32.8%)** |

**Table 3. Significant factors associated with COVID-19 vaccination.**

| | aOR (CI 95%) | *P*-value |
|---|---|---|
| **Age 55 years or older (vs. 18–55)** | 1.74 (1.3–2.34) | < .001 |
| **Married (vs. not married)** | 1.48 (1.22–1.79) | < .001 |
| **Previously tested for COVID-19 (Yes vs. No)** | 1.23 (1.05–1.43) | .039 |
| **Been tested for COVID-19 (Yes vs. No)** | 1.26 (1.07–1.48) | < .001 |
| **Rural resident (vs. Urban)** | 2.29 (1.77–2.96) | < .001 |
| **Perception of risk (Yes vs. No)** | 1.84 (1.28–2.64 | < .001 |
| **Has received other adult non-COVID-19 vaccinations (Yes vs. No)** | 1.77 (1.50–2.09) | < .001 |
| **Would take a COVID-19 vaccine if they had one available in their province/commune/neighborhood or village/routine vaccination sites (Yes vs. No)** | 2.22 (1.74–2.83) | < .001 |
| **Has a vaccination site within their health structure (Yes vs. No)** | 3.05 (2.51–3.69) | < .001 |
| **Has ever worked in a COVID-19 vaccination site (Yes vs. No)** | 1.87 (1.56–2.24) | < .001 |
| **Would take a COVID-19 vaccine if they knew that several vaccines against COVID-19 are present in their province/commune/district or village/vaccination sites (Yes vs. No)** | 1.55 (1.11–2.17) | .011 |

Outputs are from a multivariable logistic regression model fitted using the step-by-step Wald method.

One demographic factor that was significant in the univariable analyses (but not in the multivariable logistic regression) was gender, with respondents identifying as male more likely to report being vaccinated (OR = 1.46; 95% CI = 1.30–1.62; p < .001) (S1 Table). Likewise, the univariable regression suggested an association between having a known chronic illness and being vaccinated (OR = 1.51; 95% CI = 1.29–1. 81; p < .001).

Respondents were also asked about the factors that influenced them to either accept vaccination or not. Individuals who had received at least one vaccine could select one or more motivating factors from a list. The most frequently selected motivation, representing almost half of all selected responses, was "to protect myself and protect others" (Table 4). This was also the

**Table 4. Motivation factors for uptake, among vaccinated respondents, by province.**

| | Haut Katanga (%) | Kasai Oriental (%) | Kinshasa (%) | Kongo Central (%) | Lualaba (%) | North Kivu (%) | South Kivu[a] (%) | Total times response selected (% of total responses) |
|---|---|---|---|---|---|---|---|---|
| **To protect myself and protect others** | 345 (41.9%) | 367 (39.4%) | 358 (38.1%) | 320 (45.7%) | 188 (39.1%) | 137 (51.9%) | 432 (92.7%) | **2147 (46.61%)** |
| **To help stop transmission of the virus** | 156 (19.0%) | 249 (26.7%) | 234 (24.9%) | 183 (26.1%) | 132 (27.4%) | 50 (19.0%) | 15 (3.2%) | 1019 (22.21%) |
| **Belief in vaccination and science** | 79 (9.6%) | 113 (12.1%) | 163 (17.3%) | 75 (10.7%) | 21 (4.4%) | 40 (15.2%) | 6 (1.3%) | **497 (10.79%)** |
| **To facilitate own travel** | 103 (12.5%) | 86 (9.2%) | 82 (8.7%) | 34 (4.9%) | 47 (9.8%) | 11 (4.2%) | 10 (2.1%) | **373 (8.10%)** |
| **To return to "normal" life without restrictions** | 96 (11.7%) | 42 (4.5%) | 57 (6.1%) | 63 (9.0%) | 70 (14.6%) | 12 (4.5%) | 2 (0.4%) | **342 (7.43%)** |
| **To not die** | 43 (5.2%) | 70 (7.5%) | 43 (4.6%) | 16 (2.3%) | 18 (3.7%) | 14 (5.3%) | N/A* | **204 (4.42%)** |
| **Other specified reasons** | 1 (0.001%) | 5 (0.5%) | 3 (0.3%) | 9 (1.3%) | 5 (1.0%) | 0 (0.0%) | 1 (0.2%) | **24 (0.52%)** |
| **Total responses per province (% of total responses)** | **823 (17.87%)** | **932 (20.23%)** | **940 (20.41%)** | **700 (15.20%)** | **481 (10.44%)** | **264 (5.73%)** | **466 (10.12%)** | **4,606 (100%)** |

Respondents were able to select more than one response.

[a]South Kivu respondents were requested to only select one primary motivation. "To Not Die" was not listed as an option in South Kivu.

**Table 5. Primary reason for refusal of the COVID-19 vaccine, among unvaccinated respondents, by province.**

| | Haut Katanga (%) | Kasaï Oriental (%) | Kinshasa (%) | Kongo Central (%) | Lualaba (%) | North Kivu (%) | South Kivu (%) | Total times selected as primary motivation (% of total responses) |
|---|---|---|---|---|---|---|---|---|
| Insufficient data on the safety of the new vaccine | 162 (34.1%) | 49 (34.5%) | 59 (11.4%) | 110 (20.1%) | 12 (3.7%) | 64 (22.5%) | 112 (25.3%) | **568 (20.8%)** |
| Concern regarding vaccine ineffectiveness | 86 (18.1%) | 24 (16.9%) | 129 (25.0%) | 97 (17.7%) | 55 (17.1%) | 71 (25.0%) | 87 (19.7%) | **549 (20.1%)** |
| Concern regarding vaccine side effects | 72 (15.2%) | 25 (17.6%) | 173 (33.5%) | 71 (13.0%) | 39 (12.1%) | 113 (39.8%) | 53 (12.0%) | **546 (20.0%)** |
| I am against vaccines in general | 34 (7.2%) | 2 (1.4%) | 63 (12.2%) | 113 (20.6%) | 31 (9.7%) | 3 (1.1%) | 45 (10.2%) | **291 (10.7%)** |
| Lack of trust because of the short time frame to manufacture vaccines | 51 (1.1%) | 5 (3.5%) | 29 (5.6%) | 21 (3.8%) | 95 (29.6%) | 1 (0.4%) | 8 (1.8%) | **210 (7.7%)** |
| God's protection is enough, there is no need for a vaccine | 13 (2.7%) | 18 (12.7%) | 17 (3.3%) | 23 (4.2%) | 48 (15.0%) | 4 (1.4%) | 52 (11.8%) | **175 (6.4%)** |
| Because of the Westerners or Illuminati's plan to eliminate the Africans through vaccines | 40 (8.4%) | 3 (2.1%) | 4 (0.8%) | 29 (5.3%) | 26 (8.1%) | 2 (0.7%) | 25 (5.7%) | **129 (4.7%)** |
| Previous adverse reaction to any vaccine | 3 (0.6%) | 9 (6.3%) | 21 (4.1%) | 26 (4.7%) | 8 (2.5%) | 14 (4.9%) | 24 (5.4%) | **105 (3.8%)** |
| Concern about acquiring COVID-19 infection from the vaccine itself | 3 (0.6%) | 2 (1.4%) | 2 (0.4%) | 23 (4.2%) | 1 (0.3%) | 3 (1.1%) | 14 (3.2%) | **48 (1.8%)** |
| I do not perceive myself to be at high risk for COVID-19 infection | 4 (0.8%) | 4 (2.8%) | 6 (1.2%) | 14 (2.6%) | 1 (0.3%) | 3 (1.1%) | 11 (2.5%) | **43 (1.6%)** |
| I perceive myself as not being at considerable risk of developing complications if I am infected with COVID-19 | 6 (1.3%) | 1 (0.7%) | 7 (1.4%) | 8 (1.5%) | 1 (0.3%) | 3 (1.1%) | 10 (2.3%) | **36 (1.3%)** |
| Vaccine administration is painful or inconvenient. | 0 (0%) | 0 (0%) | 3 (0.6%) | 13 (2.4%) | 2 (0.6%) | 2 (0.7%) | 0 (0%) | **20 (0.7%)** |
| I have already had an infection with COVID-19 | 1 (0.2%) | 0 (0%) | 3 (0.6%) | 0 (0%) | 2 (0.6%) | 1 (0.4%) | 1 (0.2%) | **8 (0.3%)** |
| **Total responses (% of total responses)** | **475 (17.4%)** | **142 (5.2%)** | **516 (23.6%)** | **548 (25.2%)** | **321 (14.7%)** | **284 (13.0%)** | **442 (20.3%)** | **2,728 (100%)** |

most frequently selected single response, which was chosen as the only motivating factor by 32.1% of respondents. "To stop transmission of the virus" was the second most frequently selected motivating factor overall, although it represented fewer than a quarter of all responses. These also were the top two most frequently selected motivations in each province, although the third most frequently selected motivation varied between "belief in vaccination and science" (in Kasai Oriental, Kinshasa, Kongo Central and North Kivu), "to facilitate own travel" (Haut Katanga and South Kivu), and "to return to 'normal' life without restrictions" (Lualaba).

Nonvaccinated respondents were asked to select their primary reason for refusal, out of a pre-prepared list of potential responses. Overall, perceived insufficiency of data over vaccine safety was the most frequently cited reason for refusal and also the most frequently selected in Haut Katanga, Kasaï Oriental, and South Kivu provinces (Table 5). The second most frequently cited reason overall was concern regarding vaccine ineffectiveness, although this did not appear as the top reason in any individual province. Instead, other most frequently cited reasons at the provincial level were concerns over vaccine side effects (Kinshasa and North Kivu), lack of trust because of the short time frame for manufacture of the vaccines (Lualaba), and being against vaccines in general (Kongo Central). Reasons relating to perceived lower risk of infection or complications with COVID-19, or existing natural immunity through infection, were among the least frequently selected responses overall and within each province.

Both vaccinated and unvaccinated respondents were asked to provide their level of agreement with a series of statements related to factors that might influence or incentivize them to receive a COVID-19 vaccination, and to other statements related to trust and effectiveness of the national COVID-19 response. Vaccinated respondents had significantly higher agreement levels with every provided influencing factor compared with unvaccinated individuals (S2 Table). However, both groups had the strongest agreement rates with the same two factors: "If I were convinced that getting vaccinated would help protect vulnerable members of my family or community" (vaccinated respondents: mean = 3.21, standard deviation [SD] = 1.36; unvaccinated respondents: mean = 2.62, SD = 1.47) and "If I were sure that the vaccine is effective and that people who are vaccinated do not get sick with COVID-19" (vaccinated respondents: mean = 3.04, SD = 1.41; unvaccinated respondents: mean = 2.75, SD = 1.50). Both groups also had the same statements with which they agreed the least, related to receiving food or money as incentives for getting vaccinated (vaccinated respondents: mean = 1.66 and 1.64; SD = 1.00 and 1.04, respectively; unvaccinated respondents: mean = 1.46 for both, SD = 0.84 and 0.88, respectively).

Vaccinated respondents had higher agreement with all the statements related to trust in the authorities, media, health system, and government actions and measures related to the COVID-19 response (S3 Table).

## Discussion

Vaccination is the most effective method of averting vaccine-preventable diseases. However, vaccine hesitancy can compromise vaccination considerably [22], and lack of uptake in HCWs, who are at elevated risk for occupational exposure to diseases like COVID-19 [23], is particularly important for health systems resilience during epidemics.

The overall percentage of individuals who had received at least one dose of COVID-19 vaccine among the over 5,100 HCWs surveyed was 46.3%, which was similar to the level observed for the continent of Africa as a whole, in a 2022 meta-analysis [24]. However, that study showed an overall rate of acceptance in Central Africa of 28%, which could suggest higher rates of acceptance in DRC, or perhaps a temporal change in acceptability. Overall, vaccine hesitancy among HCWs in DRC, and Africa as a whole, is higher than in other regions of the world; one scoping review found acceptance in HCWs of over 75% globally [25]. Our survey findings suggest that in DRC, despite being higher than expected for the region, the uptake is much less than targets; for example, the WHO suggests that countries should aim for a vaccination rate of 100% of HCWs to achieve 70% coverage of the overall population [26].

The main reasons for vaccine hesitancy among HCWs in this study are related to safety, side effects, and effectiveness. This aligns with other findings, including a scoping review of 12 studies, that the reasons for vaccine hesitancy in all studies cited safety, side effects, or adverse events [24]. Additional factors cited in other studies, but which were less predominant motivations for vaccine hesitancy among the DRC HCWs surveyed here, included the short duration of clinical trials, lack of trust in the vaccine sources, the low severity of COVID-19, and the risk of acquiring COVID-19 from the vaccine [24,27–29]. We also observed that younger individuals were less likely to be vaccinated, and in the univariable analysis, women were also significantly more hesitant than men, findings also seen in studies among HCWs in Ghana and Ethiopia, for example, as well as globally [28,30–34]. Regarding age, it is possible that older HCWs are more aware of the strong link between age and severe COVID-19 outcomes; however, we did not find a clear association between self-perceived risk of contracting COVID-19 and vaccination status, let alone stratified by age. Taken together, the combination of younger women being more hesitant could suggest that misinformation, specifically around side effects

relating to infertility or other reproductive impacts, is affecting uptake among HCWs in DRC, which would mirror observations from other studies and settings [35–37].

Our study also showed that place of residence was significantly associated with vaccine uptake, with rural populations more likely to be vaccinated. However, our sample was skewed quite heavily toward urban residents, as we specifically targeted urban healthcare facilities for distribution of the survey. Our findings contrast with those from other countries, such as India and the United States, where rural communities are consistently more vaccine-hesitant than urban populations [34,38,39]. However, in DRC and other African settings, rural populations are less likely to use mobile phones and to use them to access the internet compared with urban dwellers [40,41]. Given the highly impactful role of social media in spreading misinformation, it could mean that in countries like DRC, HCWs and the general population in urban settings are more exposed to media that might contribute to hesitancy.

Vaccinated individuals in this study described their primary motivations for receiving a vaccine as being predominantly to protect themselves from disease and to protect their friends, family, and loved ones, a finding mirrored in other studies of vaccine uptake in Africa [42]. Vaccinated respondents also noted that they would be willing to take vaccines if they are provided in their health structure or local area; which, together with additional messaging emphasizing the positive impact of vaccination on family and community, could be a helpful strategy to promote completion of vaccination courses or uptake of boosters.

While vaccinated individuals reported significantly higher trust scores in government, the health authorities, and other actions of the COVID-19 response, the absolute values were still quite low. This suggests that the government could increase efforts to build trust among this key population, especially with respect to preparedness for future epidemics, as trust has been shown to be a key factor in promoting compliance to response measures, and lower mortality outcomes, during health emergencies [43,44].

We did not observe any factors that would strongly motivate unvaccinated individuals to receive vaccine doses. As seen in other settings, vaccine-hesitant HCWs may therefore benefit from tailored messaging to assuage concerns related to safety and side effects in particular, while also attempting to build trust [32]. Future research could also aim to investigate trusted sources of information among vaccine-hesitant HCWs and leverage those channels for more targeted communication approaches.

This study had several limitations. First, the method of sampling health facilities in proximity to the capital cities of the provinces may lead to results that are not generalizable to HCWs throughout the province. Second, administration of the questionnaire by data collectors instead of through an anonymous method, may have led respondents to respond less accurately or honestly about their vaccination status and beliefs and practices related to vaccination. Third, the KAP questionnaires were administered in the seven provinces between December 2021 and November 2022, during which time the government's vaccination campaigns continued to roll out. Consequently, surveying HCWs in provinces that were in different phases of vaccine rollout may have contributed to differences in vaccine uptake.

## Conclusion

Hesitancy to vaccinate against COVID-19 among health professionals may have a negative impact on progress to build public confidence in the COVID-19 vaccination program. Our results suggest the need to develop tailored strategies to address the concerns identified in the study to ensure optimal vaccine acceptance among HCWs in DRC. Future research, which should include qualitative data collection, should seek to understand specific concerns with

respect to side effects and safety in unvaccinated individuals to inform the development of more targeted vaccination messaging.

## Supporting information

**S1 Table. Univariate regression analyses of factors significantly associated with COVID-19 vaccine status (N = 5,102).** Percentages are calculated across rows. Reference variable noted in the OR column. OR = odds ratio; CI = confidence interval.
(DOCX)

**S2 Table. Decisions influencing COVID-19 vaccination, by vaccination status.** Responses were recorded on a Likert scale, with 1 the least level of agreement and 5 the strongest level of agreement. A response of 3 was described as "partially agree".
(DOCX)

**S3 Table. Participant's level of confidence and social trust in government authorities in the fight against COVID-19, by vaccination status.** Responses were recorded on a Likert scale, with 1 the least level of agreement and 5 the strongest level of agreement. A response of 3 was described as "partially agree".
(DOCX)

**S1 File. Knowledge, Attitudes, Practices Questionnaire (in French and English).**
(DOCX)

**S2 File. Dataset of KAP questionnaire responses.**
(CSV)

## Acknowledgments

We would like to thank all the healthcare workers who contributed their time and perspectives to this study, the data collectors who administered the surveys, as well as the support of the Ministry of Health and other COVID-19 response partners. Additionally, we would like to thank colleagues from the US Centers for Disease Control and Prevention for their input into the design of the study: Brooke Aksnes, Melissa Dahlke, Reena H. Doshi, Norbert Soke Gnakub, Richard Luce, and Robert Perry. This publication was supported by Cooperative Agreement NU2HGH000047 funded by the US Centers for Disease Control and Prevention.

## Author Contributions

**Conceptualization:** Michel K. Nzaji, Christophe Luhata Lungayo, Aime Cikomola Mwana Bene, Anselme Manyong Kapit, Pia D. M. MacDonald, Kristen B. Stolka.

**Data curation:** Michel K. Nzaji, Shanice Fezeu Meyou, Dana Sessoms, Claire J. Standley, Kristen B. Stolka.

**Formal analysis:** Michel K. Nzaji, Jean de Dieu Kamenga, Shanice Fezeu Meyou, Alanna S. Fogarty, Dana Sessoms, Claire J. Standley.

**Funding acquisition:** Anselme Manyong Kapit, Pia D. M. MacDonald, Kristen B. Stolka.

**Investigation:** Michel K. Nzaji, Jean de Dieu Kamenga, Christophe Luhata Lungayo, Aime Cikomola Mwana Bene, Shanice Fezeu Meyou, Anselme Manyong Kapit, Kristen B. Stolka.

**Methodology:** Michel K. Nzaji, Jean de Dieu Kamenga, Anselme Manyong Kapit, Pia D. M. MacDonald, Kristen B. Stolka.

**Project administration:** Shanice Fezeu Meyou, Anselme Manyong Kapit, Pia D. M. MacDonald, Kristen B. Stolka.

**Resources:** Anselme Manyong Kapit.

**Supervision:** Christophe Luhata Lungayo, Anselme Manyong Kapit, Pia D. M. MacDonald, Claire J. Standley, Kristen B. Stolka.

**Validation:** Michel K. Nzaji, Jean de Dieu Kamenga, Christophe Luhata Lungayo, Aime Cikomola Mwana Bene, Anselme Manyong Kapit, Alanna S. Fogarty, Claire J. Standley, Kristen B. Stolka.

**Visualization:** Michel K. Nzaji, Shanice Fezeu Meyou.

**Writing – original draft:** Michel K. Nzaji, Pia D. M. MacDonald, Claire J. Standley, Kristen B. Stolka.

**Writing – review & editing:** Michel K. Nzaji, Jean de Dieu Kamenga, Christophe Luhata Lungayo, Aime Cikomola Mwana Bene, Shanice Fezeu Meyou, Anselme Manyong Kapit, Alanna S. Fogarty, Dana Sessoms, Pia D. M. MacDonald, Claire J. Standley, Kristen B. Stolka.

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
