## [Decision Letter · Decision Letter 0]

18 Dec 2023

Factors associated with COVID-19 vaccine uptake and hesitancy among healthcare workers in the Democratic Republic of the Congo

PGPH-D-23-01870

Dear Dr. Standley,

We are pleased to inform you that your manuscript 'Factors associated with COVID-19 vaccine uptake and hesitancy among healthcare workers in the Democratic Republic of the Congo' has been provisionally accepted for publication in PLOS Global Public Health.

While the reviewers had a few minor comments for readability (see below), these are not essential for this manuscript to meet publication criteria and will leave it to the authors to decide if they choose to adopt these minor suggestions prior to publication. 

Best regards,

Sarah E. Brewer, PhD

Academic Editor

Reviewer Comments (if any, and for reference):

Reviewer's Responses to Questions

**Comments to the Author**

1. Does this manuscript meet PLOS Global Public Health’s publication criteria? Is the manuscript technically sound, and do the data support the conclusions? The manuscript must describe methodologically and ethically rigorous research with conclusions that are appropriately drawn based on the data presented.

Reviewer #1: Yes

Reviewer #2: Yes

Reviewer #3: Yes

2. Has the statistical analysis been performed appropriately and rigorously?

Reviewer #1: Yes

Reviewer #2: Yes

Reviewer #3: Yes

3. Have the authors made all data underlying the findings in their manuscript fully available (please refer to the Data Availability Statement at the start of the manuscript PDF file)?

Reviewer #1: Yes

Reviewer #2: Yes

Reviewer #3: Yes

4. Is the manuscript presented in an intelligible fashion and written in standard English?

Reviewer #1: Yes

Reviewer #2: Yes

Reviewer #3: Yes

5. Review Comments to the Author

Reviewer #1: This manuscript assessed the factors associated with COVID-19 vaccine uptake and hesitancy among healthcare workers in several regions in the Democratic Republic of the Congo.

Well conceptualized and structured write-up

Specific comments:

Page 5, Line 93: Kindly check the format of the dates and make it consistent throughout the manuscript.

Page 17, Line 230: "preventing vaccine-preventable diseases" - Kindly revise; there is some repetition here.

Reviewer #2: The authors have demonstrated a strong knowledge of the subject matter and written the paper very well. The abstract was consistent with the guidelines of the journal, the main text was adequate and reads very well, the methods was thorough and reproducible, the results were detailed and chronologically presented, and the discussion was also well presented. The discussion leads the reader straight to the key issues underpinning the study. Given the public relevance of the topic, it is hoped that health managers in Africa, especially, will find the findings worthy of consideration. The study touched on a key issue that dominated the fight against Covid-19 in the world, but more so in Africa. It is my expectation that health managers in Africa, especially, may take interest in the findings of this paper to help prepare better for future outbreaks. Moreover, the lessons would also be useful for dealing with other vaccine-based diseases. The findings were pretty revealing. I congratulate the authors for a good job done. However, I suggest that future studies, the authors should concentrate more on their findings and less of other relevant studies. This does not undermine the quality of the paper in anyway at all, I must say.

Reviewer #3: Covid-19 Vaccine uptake and hesitancy are seen as critical factors in the success or failure of the global COVID-19 response. This well researched and written paper examines these factors in a unique population- health care workers in a low-income country in Africa. The paper also discusses the possible rationale for both uptake and hesitancy. I would suggest some minor revisions to improve readability.

ABSTRACT:

Lines 39 to 41: Revise this opening sentence in the Conclusion to: "The findings suggest an opinion divide between HCWs willing to be vaccinated and HCWs who are hesitant to be vaccinated. A multidimensional approach may be needed to increase the acceptability of the COVID-19 vaccine for HCWs."

6. PLOS authors have the option to publish the peer review history of their article (what does this mean?). If published, this will include your full peer review and any attached files.

**Do you want your identity to be public for this peer review?** For information about this choice, including consent withdrawal, please see our Privacy Policy.

Reviewer #1: **Yes: **Prof. Joseph Fadare

Reviewer #2: **Yes: **BOTHA, Nkosi Nkosi

Reviewer #3: **Yes: **Paul R De Lay, MD, DTM&H (Lond(
